# *Gbx2* Is Required for the Migration and Survival of a Subpopulation of Trigeminal Cranial Neural Crest Cells

**DOI:** 10.3390/jdb8040033

**Published:** 2020-12-11

**Authors:** David A. Roeseler, Lona Strader, Matthew J. Anderson, Samuel T. Waters

**Affiliations:** 1MilliporeSigma, St. Louis, MO 63103, USA; david.roeseler@milliporesigma.com; 2Division of Sciences and Mathematics, University of the District of Columbia, Washington, DC 20008, USA; Lstrader824@gmail.com; 3Cancer and Developmental Biology Laboratory, National Cancer Institute, Frederick, MD 21702, USA; andersonmj@mail.nih.gov

**Keywords:** *Gbx2*, anterior hindbrain, neural crest, trigeminal ganglion, mouse, development

## Abstract

The development of key structures within the mature vertebrate hindbrain requires the migration of neural crest (NC) cells and motor neurons to their appropriate target sites. Functional analyses in multiple species have revealed a requirement for the transcription factor gastrulation-brain-homeobox 2 (Gbx2) in NC cell migration and positioning of motor neurons in the developing hindbrain. In addition, loss of *Gbx2* function studies in mutant mouse embryos, *Gbx2^neo^*, demonstrate a requirement for *Gbx2* for the development of NC-derived sensory neurons and axons constituting the mandibular branch of the trigeminal nerve (CNV). Our recent GBX2 target gene identification study identified multiple genes required for the migration and survival of NC cells (e.g., *Robo1, Slit3, Nrp1*). In this report, we performed loss-of-function analyses using *Gbx2^neo^* mutant embryos, to improve our understanding of the molecular and genetic mechanisms regulated by *Gbx2* during anterior hindbrain and CNV development. Analysis of *Tbx20* expression in the hindbrain of *Gbx2^neo^* homozygotes revealed a severely truncated rhombomere (r)2. Our data also provide evidence demonstrating a requirement for *Gbx2* in the temporal regulation of *Krox20* expression in r3. Lastly, we show that *Gbx2* is required for the expression of *Nrp1* in a subpopulation of trigeminal NC cells, and correct migration and survival of cranial NC cells that populate the trigeminal ganglion. Taken together, these findings provide additional insight into molecular and genetic mechanisms regulated by *Gbx2* that underlie NC migration, trigeminal ganglion assembly, and, more broadly, anterior hindbrain development.

## 1. Introduction

The wide range of phenotypes reported in *Gbx2* mutant organisms suggest that *Gbx2* contributes to numerous divergent cell types and their derivatives during development [1,2,3,4]. The highly motile cranial neural crest (NC) cells are one such cell type that *Gbx2* is thought to regulate during vertebrate development [5,6]. Multipotent cranial NC cells delaminate and migrate ventrolaterally from the dorsal neuroepithelium, forming components of the cranial ganglion and craniofacial skeletal structures in the distal branchial arches [7]. Cranial NC cells from the midbrain will innervate the distal region of branchial arch (BA)1 and give rise to components of the upper jaw bones and cartilage, while the anterior hindbrain cranial NC cell populations innervate the proximal region to form elements of the lower jaw [8].

The trigeminal cranial ganglia (nV) contains placodal and cranial NC cell-derived sensory neurons as well as motor neurons originating from the ventral neural tube. Both sensory and motor neurons project their axons into BA1 to innervate muscles in the jaw and facilitate mastication [9]. Accumulating data suggest that enzymes that regulate cellular retinoic acid (RA) levels, an integrative network of transcription factors, and secreted signaling molecules are required for nV progenitors to acquire and maintain their correct spatial identities. Retinoic acid (RA) plays important roles in vertebrate embryonic development and is indispensable in anteroposterior (A-P) patterning of the central nervous system. In addition, the RA-degrading enzymes, CYP26A1 and CYP26C1, are thought to partially regulate the posteriorizing effects of RA in the anterior hindbrain and play a significant role in the production of migratory cranial NC cells [10].

Loss-of-function studies for several transcription factors expressed in the developing anterior hindbrain in mice (e.g., *Hoxa2*, *Hoxb2*, and *Krox20*) have demonstrated a significant role for these molecules in patterning this region by regulating the spatial molecular identity of transverse developmental units, rhombomeres (r). *Hox* paralog group 2 genes, *Hoxa2* and *Hoxb1*, function to control the anteroposterior (A–P) and dorsoventral (D–V) patterning of neuronal subtypes in the anterior hindbrain [11,12].

*Krox20*, encoding a zinc finger transcription factor, is expressed before the occurrence of sharp segmental boundaries in the anterior hindbrain and eventually becomes restricted to and identifies r3 and r5. *Krox20* expression is seen in r5 neural crest cells at E8.5 and begins to downregulate in r3 at E9.5. [13]. By E10.5, *Krox20* expression specifically identifies the nV and nVII neural crest-derived boundary cap (BC) cells, demarcating the entry and exit points for the connecting axons from the central nervous systems to the periphery [14,15]. Loss-of-function studies for *Krox20* demonstrate that r3 and r5 cells mix with and acquire r2 and r4 phenotypic characteristics and further exhibit a loss of direct target genes, *Hoxa2* and the receptor tyrosine kinase gene, *Epha4* [14,16,17]. The loss of odd-numbered rhombomere identity in *Krox20*^−/−^ mutants alters the ensuing gene expression patterns in hindbrain neurons, cranial neurons, and cranial NC derivatives [18].

Most functional studies regarding *Gbx2* have highlighted its role in defining the midbrain–hindbrain boundary (MHB) organizer and development of the anterior hindbrain. Gene inactivation studies in mice have shown that expression of *Gbx2* is required to maintain normal patterns of expression of key components of MHB organizer, *Fgf8* and *Wnt1*, and correct formation of r1–r3 and their derivatives, such as the cerebellum and nV [2,19,20,21,22]. 

Importantly, loss of *Gbx2* function in mouse embryos (*Gbx2*^−/−^) results in aberrant NC cell patterning [3,5]. Misexpression studies have demonstrated a requirement for *Gbx2* for the correct development of NC-derived structures. During mouse development between E8.5 and E9.5, *Gbx2* is expressed in the dorsal anterior hindbrain, corresponding to the location where cranial NC cells originate, and persists throughout the major period of NC migration [5,19,23]. Studies in *Gbx2*^−/−^ mutants have shown a requirement for *Gbx2* in NC cell migration. These studies identified defects in migratory cranial NC cells, leading to defects in NC derivatives in the central nervous system and abnormalities in craniofacial and cardiovascular components [2,5,6]. Disruption in *Slit*/*Robo* signaling and loss of GBX2 target gene *Robo1* expression are thought to partially contribute to perturbations within the migratory r4 and cardiac NC cell populations and contribute to defects observed with the anterior hindbrain and heart [5,6].

In addition to *Gbx2*^−/−^, a *Gbx2* hypomorph mouse has also been characterized. Mouse embryos homozygous for the hypomorphic allele, *Gbx2^neo^*^/*neo*^, only express 6–10% of wild-type *Gbx2*. As a result, r1 and r3 are present while r2 is significantly reduced in *Gbx2^neo^*^/*neo*^ mutant embryos. In situ hybridization analyses of *Gbx2^neo^*^/*neo*^ mice revealed a loss of genes expressed in r2 from E8.5 to E9.5, including *Cyp26c1* and *Hoxa2.* Disruption of r2 significantly impacts components derived from this region. These defects include loss of nV cranial NC-derived sensory neurons, the nV mandibular branch, and the nV NC-derived BC cells connecting the peripheral and central neural circuitry. The reduction of r2 and the loss of cranial NC cell-derived components in *Gbx2^neo^*^/*neo*^ mutant embryos support a stringent requirement for *Gbx2* in the regulation of migrating cranial NC cells and are thought to contribute to early postnatal death [24].

Since a large portion of the anterior hindbrain, r1–r3, fails to develop in *Gbx2*^−/−^ embryos, it makes them inappropriate for analyses of many of the anterior hindbrain abnormalities associated with a loss of *Gbx2* function. In this study, we utilized *Gbx2^neo^*^/*neo*^ embryos and loss-of-function analyses to improve our understanding of the molecular and genetic mechanisms regulated by *Gbx2* during anterior hindbrain and nV development. We report a significant reduction in r2-derived motor neurons and misregulation of *Krox20* expression in r3. In addition, we demonstrate a requirement for *Gbx2* for *Nrp1* expression within a subpopulation of nV NC-derived cells, cranial NC cell migration, and survival of cranial NC cells. These results provide novel insight into the multiple functions of *Gbx2* during anterior hindbrain development and expand our understanding for the role of *Gbx2* in the regulation of migrating cranial NC cells and nV gangliogenesis.

## 2. Materials and Methods

### 2.1. Mice and Genotyping

Descriptions of mice carrying the allele for *Gbx2^neo^* and the genotyping have been previously presented [2]. *Gbx2^neo^*^/*neo*^ hypomorphic mice contain all exons (1 and 2). The hypomorphic phenotype is due to the insertion of a 497 bp fragment of a neo-resistance cassette used as a selectable marker in gene-targeting, between exons 1 and 2 [24]. *Gbx2^neo^*^/*neo*^ hypomorphic mice were maintained on a mixed 129XC57BL/6 background while *Gbx2^neo^*^/*+*^ and wild-type mice served as controls.

### 2.2. In Situ Hybridization

Whole-mount in situ hybridization was performed as previously described [25]. In situ fluorescein and digoxigenin-labeled riboprobes were transcribed from pBluescript KS(−) plasmids containing the following cDNA fragments: *Nrp1* anti-sense probe was provided by Q. Schwarz. The *Sox10* anti-sense probe was provided by A. Chandrasekhar while the constructs were engineered in P. Trainor’s lab.

### 2.3. Electrophoretic Mobility Shift Assay

Recombinant GBX2 protein purification and gel shift assays were performed as previously described [5]. Synthesized oligonucleotides were annealed and end-labeled with (γ-^32^P ATP) [5]. For the gel shift assay, 940 pM of labeled probe and 720 nM of GBX2 fusion protein were mixed and incubated with 2X binding buffer (20 mM Tris-HCL, pH 7.4, 50 mM KCL, 1 mM fresh dithiothreitol, 10% glycerol, 200 µg/mL bovine serum albumin (BSA) and 2.2 µg/mL Poly dI/dC) for 30 min at 25 °C. The reactions were separated on a 6% nondenaturing polyacrylamide gel at 300 v at room temperature using a 1.0 × Tris-Glycine running buffer. The sequences for the NRP1 oligonucleotides containing the GBX2 binding sites are as follows:NRP1-F: 5′aacattccaaaaattatcaaccatttcaggaatacatttc**ataaaa**gctagattgagttctgcttgttttttatt 3′NRP1-R: 5′aataaaaaacaagcagaactcaatctagc**ttttat**gaaatgtattcctgaaatggttgataatttttggaatgtt 3′NRP1-F: mutated 5′aacattccaaaaattatcaagattgagttctgcttgttttttatt 3′NRP1-R: 5′aataaaaaacaagcagaactcaatcttgataatttttggaatgtt 3′

### 2.4. Immunohistochemistry

Mouse embryos were fixed in 4% paraformaldehyde for 2 h at 4 °C and embryos were subsequently washed three times in 1X phosphate-buffered saline-diethyl pyrocarbonate (PBS-DEPC). Embryos were incubated in 25% sucrose overnight and embedded in optimal cutting temperature (O.C.T.) Compound (Tissue-Tek Sakura Finetek USA, Inc., Torrance, USA). Serial transverse 12 μm cryosections were dried for 60 min at room temperature (RT) and were washed in PBST (1X PBS-DEPC and 1% Triton X-100). Sections were incubated with blocking solution (10% heat-inactivated lamb serum, 1% BSA, 2.5% Triton X-100) for 90 min at RT and incubated with the appropriate primary antibodies overnight at 4 °C at the given dilutions: mouse anti-AP-2 alpha (1:25; 3B5 DSHB), rabbit anti-cleaved caspase-3 (1:1600; Cell Signaling Technology, Danvers, USA), and rabbit anti-E-cadherin (1:100; 24E10 Cell Signaling Technology, Danvers, USA). The sections were washed three times the following day with PBST and incubated with fluorescently conjugated secondary antibodies at the given dilutions: goat anti-rabbit AlexaFluor 488 and goat anti-mouse Alexafluor 568 (1:500; Invitrogen, Pittsburgh, PA, USA). Stained sections were washed six times in PBST and mounted with Vectashield Mounting Media with 4’,6’-diamidino-2-phenylinodole (DAPI) (Vector BioLabs, Malvern, PA, USA).

### 2.5. Statistical Analysis

Statistical differences in Casp3-positive apoptotic cells were assessed using Student’s *t*-test (Graphpad Prism software, GraphPad, San Diego, USA). Results are represented as mean ± SEM, and samples are considered statistically significant by having a value of * *p* ≤ 0.0002. *n* = 3 WT embryos, *n* = 3 *Gbx2^neo^*^/*neo*^ embryos.

### 2.6. Animal Ethics Statement

Animal experimentation protocols were reviewed and approved by the University of Missouri IACUC (protocol #7561).

## 3. Results

### 3.1. Loss of Gbx2 Results in an Inability to Suckle and Trigeminal Motor Neuron Defects

Given the severity of anterior hindbrain defects observed in *Gbx2*^−/−^ mouse, we utilized *Gbx2^neo^*^/*neo*^ mice to investigate the motor neuron and NC cell populations that populate nV. As previously reported, analysis of *Gbx2^neo^*^/*neo*^ mice at P0 revealed that they are born with normal Mendelian frequency [2]. However, further investigation of these mice at P0 revealed a lack of colostrum in the stomachs of *Gbx2^neo^*^/*neo*^ mice compared to controls (Figure 1A,B). The absence of colostrum was observed in 100% of the *Gbx2^neo^*^/*neo*^ P0 mice examined (*n* = 10/10) and is strongly suggestive of suckling defects in *Gbx2^neo^*^/*neo*^ mice.

Trigeminal motor neurons of the nV cranial nerve arise from r2 and r3 and subsequently undergo dorsolateral migration from r2 to form the trigeminal nucleus [26]. Previous investigation of *Gbx2^neo^*^/*neo*^ mice revealed a loss of genes expressed in r2 from E8.5 to E9.5, including *Cyp26c1* and *Hoxa2* [24]. Loss-of-function studies in *Hoxa2*^−/−^ embryos have shown that *Hoxa2* is required for maintaining the molecular identity of r2 and r3 and for the correct motor neuron axon pathfinding from these regions [27]. Since *Gbx2* is essential for normal r2 development and *Hoxa2* expression is absent in r2 of *Gbx2^neo^*^/*neo*^ embryos, we examined the potential impact of reduced *Gbx2* function on motor neuron cell bodies in the anterior hindbrain. We first analyzed the development of r3 in *Gbx2^neo^*^/*neo*^ embryos using in situ analyses. At E8.5, *Krox20* is expressed in r3 and r5 in wild-type embryos (Figure 2A). Our results reveal a decrease in *Krox20* expression in the lateral domains of r3 in *Gbx2^neo^*^/*neo*^ embryos when compared to wild-type (Figure 2A,B). Interestingly, these results are consistent with our studies in zebrafish, in which *Krox20* expression in r3 is significantly reduced in *gbx2* morphants [4].

We next used in situ hybridization to analyze the expression of the T-box transcription factor *Tbx20. Tbx20* is expressed by branchiomotor neurons in the anterior hindbrain at E9.5 and E10.5 [28]. In wild-type control embryos, *Tbx20*-expressing motor neurons were observed in their ventral progenitor domain in r2–r5 at E9.5 and were beginning to migrate dorsolaterally from r2 by E10.5 (Figure 2C,E). In contrast to our previous report showing loss of r2 gene expression in *Gbx2^neo^*^/*neo*^ embryos [24], we observed a truncated population of *Tbx20*-expressing motor neurons in r2 at E9.5 and E10.5 (Figure 2D,F). To further detail our expression analysis, we performed two-color in situ hybridization analyses of *Tbx20* and *Krox20* in E9.5. Consistent with previous reports [12], we observed a downregulation of *Krox20* expression in wild-type embryos at E9.5 (Figure 2G). Surprisingly, our data show that ectopic *Krox20* expression persists in the dorsal neural tube of r3 in *Gbx2^neo^*^/*neo*^ embryos through E9.5 (*n* = 4/4) (Figure 2H). 

### 3.2. Reduction in Gbx2 Results in Trigeminal Cranial NC Cell Defects

The loss of multiple neural crest-derived structures in the anterior hindbrain of *Gbx2* mutant embryos, and the previously reported NC cell migration defects observed in *Gbx2*^−/−^ embryos, prompted us to investigate NC cell migration in *Gbx2^neo^*^/*neo*^ embryos [3,5,6]. At E9.5, cranial NC cells are observed migrating in distinct streams away from the neuroepithelium at r2 and r4, where subpopulations of NC cells will contribute to cranial ganglia, while others will migrate more ventrally into the BAs, giving rise to the bones and cartilage of the face and neck (for review, see [29]). To investigate perturbations in NC cell migration, we used in situ hybridization to analyze *Sox10* expression. *Sox10* is expressed in migrating NC cells at E9.5, and in NC cell-derived cranial sensory ganglia at E10.5 [30]. In wild-type embryos, we observed NC cells migrating from r2 into BA1 (Figure 3A,C). However, in *Gbx2^neo^*^/*neo*^ embryos, we observed a loss of NC cell migration into BA1 and a subpopulation of NC cells residing in the dorsal neuroepithelium (Figure 3B,D). Migration of NC cells from r4 into BA2 did not appear to be affected by the loss of *Gbx2* in *Gbx2^neo^*^/*neo*^ embryos.

Loss of *Krox20* expression in r3 and r5 cells results in their acquisition of even-numbered phenotypic characteristics and subsequent modification of gene expression in the anterior hindbrain and cranial NC cell derivatives [12,18]. We observed persistent ectopic expression of *Krox20* in r3 and abnormal migration of a subpopulation of *Sox10* expressing NC cells residing in the dorsal neuroepithelium of r2 in E9.5 *Gbx2^neo^*^/*neo*^ embryos (Figure 2H Figure 3D). Therefore, we examined if ectopic *Krox20* expression in r3 cells may lead to their taking on a similar even-numbered phenotype. We used in situ hybridization to determine if the expression of *Krox20* and *Sox10* identified the same population of cells in *Gbx2^neo^*^/*neo*^ embryos [31]. Consistent with our results shown in (Figure 2), *Krox20* expression persists in r3 of *Gbx2^neo^*^/*neo*^ embryos compared to wild-type at E9.5 (Figure 3E–H); however, we did not observe overlapping expression of *Krox20* and *Sox10* at the r2/r3 boundary in *Gbx2^neo^*^/*neo*^ embryos (Figure 3F,H). Together, these results suggest that migratory cranial NC cells are produced in *Gbx2^neo^*^/*neo*^ embryos and that ectopic expression of *Krox20*-expressing cells located in r3 after E9.0 does not lead to a change in their phenotypic characteristics.

### 3.3. Loss of GBX2 Target Gene, Nrp1, Expression in a Subpopulation of Trigeminal Cranial NC Cells in Gbx2^neoneo^ Embryos

Previous studies have demonstrated that NRP1 is expressed in the nV ganglion and is required for the correct migration of cranial NC cells [32,33]. Although less severe, the defects in *Nrp1*^−/−^ mutants share similarities to cranial nerve defects observed in *Gbx2* mutant embryos, suggesting that *Gbx2* may function upstream of *Nrp1* in nV development [3,32]. Interestingly, results from our recent genome-wide ChIP-Seq analysis identified *Nrp1* as a top candidate GBX2 target gene, suggesting a possible GBX2-mediated mechanism for nV ganglion development [5].

To determine if mouse GBX2 protein binds to the GBX2 cis-regulatory *NRP1* sequence, we performed an electrophoretic mobility shift assay (EMSA) using recombinant mouse GBX2 and radiolabeled *NRP1* oligonucleotides, containing the GBX2-binding sequence. Consistent with results from our GBX2 target identification study, we observed a reduction in the mobility of two shifted complexes in lanes containing full-length GBX2 (Figure 4A, black arrows) [5].

The addition of a GBX2-specific antibody resulted in a supershift, confirming the GBX2/*NRP1* interaction, while the addition of only a truncated GBX2 protein, omitting the DNA-binding homeodomain, GBX2ΔHD, resulted in a loss of a shifted complex (Figure 4A, lanes 3 and 4). In order to demonstrate sequence specificity for GBX2 and the previously identified GBX2-binding site, we used unlabeled *NRP1* oligonucleotides, containing the GBX2-binding site at 100×, 300×, and 500× molar concentrations, and observed a progressive reduction in the observed shifted complexes (Figure 4A, lanes 5–7). The observed reduction in shifted complexes was diminished when we added unlabeled truncated *NRP1* oligonucleotides, where the GBX2-binding sequence has been omitted (Figure 4A, lanes 8–10). These results demonstrate that GBX2 binds to the previously identified *NRP1* target sequence.

We next sought to examine whether *Nrp1* expression is affected in migrating trigeminal cranial NC cells in *Gbx2^neo^*^/*neo*^ embryos [32,33]. We observed *Nrp1* expression in two distinct populations within r2 and r4 of wild-type embryos overlapping with the migratory NC cell marker *Sox10* at E9.5 (Figure 4B–G). Notably, we observed a marked reduction in *Nrp1* expression in the r2 NC cell stream in *Gbx2^neo^*^/*neo*^ embryos, while the expression of *Nrp1* in r4 appeared unaffected by the loss of *Gbx2* (Figure 4H–M). *Nrp1* expression in r4 at E9.5 is consistent with previous reports [32,34]; however, our observation of *Nrp1* expression in r2 is novel. We also observed weak *Nrp1* expression at the MHB (Figure 4B,C,F,G), which is expanded caudally through r1 in *Gbx2^neo^*^/*neo*^ embryos (Figure 4H,I,L,M). Together, these results demonstrate that GBX2 binds to the previously identified *NRP1* target sequence and supports the notion that GBX2 directly regulates the expression of *Nrp1* in a subpopulation of trigeminal cranial NC cells.

### 3.4. Loss of Gbx2 Results in an Increase in Apoptosis in Migrating Trigeminal NCC

Loss-of-function studies in zebrafish using an antisense *gbx2* morpholino resulted in an increase in cell death in r2, r3, and r5 [4]. The specific loss of r2 NC cell-derived components at E10.5 and the migratory defects observed in the trigeminal stream of NC cells at E9.5 led us to investigate whether an increase in cell death may partially explain the emergent phenotypes. We performed immunohistochemical analyses in the anterior hindbrain of wild-type and *Gbx2^neo^*^/*neo*^ embryos at E9.5, the stage at which we observed NC cell migratory defects. In wild-type and *Gbx2^neo^*^/*neo*^ embryos, we observed AP2α-positive NC cells migrating away from the dorsal neuroepithelium, populating the nV ganglion and BA1 at E9.5 (Figure 5A,B,G,H).

We next investigated apoptosis as indicated by activated caspase-3, hypothesizing that an increase in apoptosis in the migratory r2 NC cell stream might contribute to the loss of neural crest-derived structures. Activated caspase-3 was observed in the mesenchyme in wild-type and *Gbx2^neo^*^/*neo*^ embryos at E9.5 (Figure 5C,D). However, a significant increase in caspase-3-positive apoptotic cells was observed within the stream of NC cells migrating into the nV ganglion and in the surrounding mesenchyme in *Gbx2^neo^*^/*neo*^ embryos compared to wild-type (*n* = 5/7) (Figure 5I,J,M). The range of severity of cell death between *Gbx2^neo^*^/*neo*^ mutants is consistent with phenotype variability reported in mice containing alternate hypomorphic alleles encoding different genes, such as *Fgf8* [35,36]. Importantly, many of the activated caspase-3-positive cells are colocalized with AP2α-positive migratory NC cells (Figure 5K,L). Thus, an increase in cell death in cranial NC cells may explain the loss of neural crest-derived sensory components of the nV ganglion.

## 4. Discussion

In animals, the ability to feed is one of the earliest and most essential behaviors to develop. Our studies of *Gbx2^neo^*^/*neo*^ mice suggest that an inability to masticate leads to their premature death at P0. In this study, we have focused our analysis on the development of cranial NC cells and nV, both of which are critical for normal mastication [12].

### 4.1. Gbx2 Is a Critical Factor for nV Gangliogenesis

The initial formation of the nV ganglion and the establishment of functional neuronal connections with target tissues is largely regulated by the correct spatial–temporal expression of numerous intrinsic transcription factors, secreted signaling and guidance molecules. A combination of motor neurons and placodal and cranial NC cell-derived sensory neurons, originating from divergent populations of cells, migrate from their progenitor domains to form the nV ganglion. The molecular networks that determine how these divergent populations of motor and sensory neurons precisely acquire their final positions and establish axonal connections with muscles in the jaw remain unclear.

Previous studies have demonstrated a requirement for *Gbx2* in positioning the MHB boundary and the juxtaposed positions of *Wnt1* and *Otx2* in the midbrain and *Fgf8* in the anterior hindbrain [2,22]. However, the motor and sensory neurons that comprise the nV ganglion have not been investigated in detail in *Gbx2* mutants. To investigate the role of *Gbx2* during nV gangliogenesis, we analyzed the motor and cranial NC cell populations in *Gbx2^neo^*^/*neo*^ embryos prior to E10.5, when nV defects are already present. The findings from our study strongly support the idea that the loss of *Gbx2* and subsequent loss of tissue disrupts nV motor neuron development and suggest a requirement for *Gbx2* in regulating the temporal expression of *Krox20* in r3. Critically, we provide evidence that *Gbx2* is required for regulating the guidance molecule *Nrp1* and the correct migration and survival of a subpopulation of cranial NC cells that populate the nV ganglion.

### 4.2. Loss of Gbx2 and Anterior Hindbrain Tissue Disrupts r2 Motor Neuron Development

Loss-of-function studies in *Gbx2^neo^*^/*neo*^ mutants have shown a loss of cranial NC cell-derived nV sensory components; however, it was not determined if the motor neurons in the anterior hindbrain are also affected by the loss of *Gbx2*. Our analysis of motor neurons in *Gbx2^neo^*^/*neo*^ mutants revealed a marked reduction in *Tbx20* expression in r2. Loss-of-function studies in mice have demonstrated a requirement for *Gbx2* in positioning the MHB, in part by directly repressing *Otx2* expression by competing with class III POU transcription factors for an *Otx2* enhancer sequence [37]. Loss of *Gbx2* results in a caudal shift of cells within the midbrain lineage at the expense of anterior hindbrain cell types [2,22,38].

Our data suggest that r2-derived motor neurons are significantly reduced but not entirely absent in *Gbx2^neo^*^/*neo*^ mutants at E9.5. This finding is not unexpected given the loss of r2-specific gene expression and loss r2 of tissue in *Gbx2^neo^*^/*neo*^ mutants. *Hoxa2* expression is essential for generating trigeminal motor neurons and is absent in r2 in *Gbx2^neo^*^/*neo*^ mutants. Further, early patterning defects and the expansion of *Fgf8*, *Wnt1*, and *Otx2* is observed in *Gbx2^neo^*^/*neo*^ embryos prior to E9.5, at the 10 somite stage (ss), which suggests that r2 cell populations may already be misspecified [24]. Data presented in this study also suggest that the loss of r2 motor neurons is not due to a defect in cell migration as we observe a loss of r2 motor neurons at E9.5, prior to the major period of motor neuron cell migration (Figure 2). While we did not observe an increase in cell death, as indicated by activated caspase-3 in the ventral r2/r3 motor neuron progenitor domains at E9.5, it remains possible that motor neuron cell death may occur slightly earlier in development. It is also possible that there is a more stringent requirement for *Gbx2* specifically for r2 motor neurons, mediating their development through a non-cell-autonomous mechanism. A thorough investigation of a multitude of motor neuron-specific markers may further elucidate the fate of r2 motor neurons; however, such an undertaking lies beyond the scope of this study.

### 4.3. Loss of Gbx2 Alters the Temporal Expression of Krox20 in r3

While gene expression in r2 is significantly impacted in *Gbx2^neo^*^/*neo*^ embryos, *Krox20* expression in r3 at E9.0 appeared to be intact [24]. Interestingly, results from this study show that ectopic expression of *Krox20* is maintained in r3 of *Gbx2^neo^*^/*neo*^ mutants at E9.5, when expression is downregulated in wild-type mice. Studies in the chick, rat, and mouse have all demonstrated that *Krox20* expression is partially regulated by numerous developmentally important transcription factors. In the mouse, *Meis*, *Hoxb1*, and *Pbx* factors directly activate *Krox20* in r3 [39]. Adding to the complexity of regulating *Krox20* expression, gain- of and loss-of-function of *Pax6* in the rat hindbrain identified a negative feedback loop via upregulation of the *Krox20*-repressor gene, *Nab1*. The network regulating *Krox20* through *Pax6*/*Nab1* is downstream of FGF signaling [40].

*Fgf8, Fgf17*, and FGF target genes, *Spry1* and *Spry4*, expression domains are expanded caudally in *Gbx2^neo^*^/*neo*^ embryos [24]. However, evidence from previous studies suggests that they do not directly contribute to the prolonged expression of *Krox20* reported in this study. A recent study in zebrafish demonstrated that FGF signaling and *Spry4* are primarily required to modulate the timing of the onset of *Krox20* expression in r3 and r5 and not maintenance later in development [41]. This is further supported by the fact that the onset of *Krox20* expression is not affected in *Gbx2^neo^*^/*neo*^ mutants (Figure 2).

Changes in RA in the anterior hindbrain may also account for alterations in *Krox20* expression given the loss of *Cyp26c1* in r2 at E9.0 in *Gbx2^neo^*^/*neo*^ mutants [24]. RA exposure in mice has been shown to induce temporal changes in *Krox20* expression in r3, where it is initially delayed at E8.5 and appears later at E10.0, altering *Hox* gene expression and inducing a posterior transformation of r2/r3 [42]. *Gbx2* is required to mediate the repression of *Otx2* by RA as it induces *Gbx2* [19]. Furthermore, in *Gbx2*^−/−^ embryos, removal of the caudally expanded *Otx2* results in the recovery of *Krox20* expression in r3, suggesting that *Gbx2* plays a permissive—not instructive—role in r3 development [19]. In the transgenic *Hoxb1-Gbx2* mice, where *Gbx2*^−/−^ embryos ectopically expressed *Gbx2* in r4 under a mouse *Hoxb1* enhancer, *Gbx2* was able to rescue *Krox20* expression in r3 [43]. These data support the notion that *Gbx2* may partially function non-cell-autonomously during r2/r3 development, in agreement with the suggested functions of *gbx2* in zebrafish [4].

*Gbx2* expression is primarily localized to r1 and not r2/r3 in the anterior hindbrain at E9.5. The data provided in this study, along with previously published results, support a non-cell-autonomous role for *Gbx2* in regulating the temporal expression of *Krox20* in r3. As *Krox20* regulates the patterning of even/odd rhombomeres through both cell-autonomous and non-cell-autonomous mechanisms, the impact of temporally misregulating *Krox20* expression in *Gbx2^neo^*^/*neo*^ mutants remains to be determined [31].

### 4.4. Misregulation of GBX2 Target Gene, Nrp1, in Gbx2^neo/neo^ Mutants

In vertebrates, neuropilins are transmembrane glycoproteins whose signaling is in part mediated by binding to members of the class 3 semaphorins (SEMA) or the vascular endothelial growth factors (VEGF) [44]. There are two NRP receptors, NRP1 and NRP2, and they mediate VEGF or SEMA signaling through interactions with either VEGF receptors or members of the plexin (PLXN) family of transmembrane glycoproteins, respectively [45,46]. Signaling through NRP and PLXN receptors elicits both chemorepulsion- and chemoattraction-mediated responses controlling both soma and axon migration during brain development [47]. The precise combination of signal and receptor proteins is thought to partially determine the repulsive and attractive responses and establish the correct migratory patterning of neuronal cells and their axons [48,49,50].

We have previously demonstrated that GBX2 directly targets *NRP1* in human prostate cancer cells (PC-3). We now show that GBX2 binds to the identified *NRP1* target sequence by EMSA and provide in vivo data supporting the notion that *Gbx2* is required to regulate *Nrp1* in the anterior hindbrain and in a subset of trigeminal cranial NC cells (Figure 5) [5]. Beyond the few studies that have investigated the function of *Nrp1* signaling in the cerebellum during the late embryonic and early postnatal developmental stages, little data are available describing a role for *Nrp1* at the very earliest stages of the developing cerebellum primordium at E9.5 [51,52]. However, there is a growing body of knowledge describing the role of *Nrp1* during axonal and soma guidance of cranial NC cells. 

Overexpression and loss-of-function studies have shown a requirement for NRP1 in neural and heart development [53,54,55]. In addition, SEMA signaling through NRP and PLXN receptors promotes the correct guidance of cardiac NC cells through the pharyngeal arches [56,57]. In the mouse, SEMA/NRP1 signaling is thought to prevent intermingling of hyoid and trigeminal cranial NC cell populations during migration [32]. Compound *Nrp1*^−/−^; *Nrp2*^−/−^ mutants have a more severe phenotype than single mutants, suggesting that these receptors function synergistically to prevent mixing of hyoid and trigeminal cranial NC cell streams [32]. However, a recent study has suggested that *Nrp2* does not compensate for *Nrp1* in migrating cranial NC cells as siRNA knockdown of *Nrp1* in chick r4 leads to a failure of cranial NC cell entry into BA2 while the additional loss of *Nrp2* had no observable effect on r4 cranial NC cell migration [58].

Intriguingly, the trigeminal cranial NC cell migratory defect observed in *Gbx2^neo^*^/*neo*^ mutants does not coincide with the cell mixing phenotypes previously reported in *Nrp1*^−/−^ mouse embryos. An explanation for the differences in phenotypes could be partially explained by the upregulation of *Krox20* in r3 and downstream target genes such as *Epha4*, which has been shown to restrict cell mingling between r2 and r4 [17,59]. Furthermore, we did not observe a complete loss of *Nrp1* in *Gbx2^neo^*^/*neo*^ mutants. The tissue-specific transcriptional regulation of *Nrp1* may be due to the *Nrp1* GBX2 binding sequence being located within an intronic *NRP1* sequence, which may therefore function as an enhancer of *Nrp1* expression within a subpopulation of cranial NC cells [5], and data presented in this study).

Previous studies have demonstrated that GBX2 interacts with Groucho/Tle co-repressor proteins and functions as both a transcriptional repressor and activator [5,60,61]. This purported transcriptional activity of GBX2 is further supported by previous studies demonstrating the importance of transcription factors modulating spatial and temporal gene expression during development and not functioning in an all-or-none response [62,63]. It is likely that a combination of tissue-specific cofactors and the chromatin architecture also account for determining the transcriptional repression or activation of *Nrp1* during development. It is also possible that GBX2 regulation of *Nrp1* may function later in development during the migration and bundling of the proximal nV axons of the mandibular branch. However, the loss of the cranial NC cell-derived component of the mandibular branch prevents us from investigating this possibility in *Gbx2^neo^*^/*neo*^ embryos. Future studies investigating *Nrp1* signaling and the determination of the downstream pathways affected by the loss of *Nrp1* in *Gbx2* mutants will further elucidate its functional role during nV development.

### 4.5. Gbx2 Is Required for the Migration and Survival of Trigeminal Cranial Neural Crest Cells

The role of *Gbx2* in neural crest cell populations has been demonstrated in studies in multiple species. Studies in chicks have demonstrated that Wnt signaling is necessary for NC cell specification, while studies in *Xenopus* have provided evidence that Wnt activation and BMP inactivation are critical for inducing NC cells [64,65]. Recent work in *Xenopus* has identified *Gbx2* as being directly activated by Wnt/β-catenin signaling, playing an essential role in the neural crest induction pathway [1].

Multiple loss-of-function studies have since demonstrated a requirement for *Gbx2* in the migration of cardiac and cranial NC cell populations. Interestingly, we recently identified members of both the Slit/Robo and Nrp/Plxn families of guidance molecules as being direct targets of GBX2. Studies in *Gbx2*^−/−^ mutants have suggested a requirement for *Gbx2*, *Robo1*, and *Slit2* for cranial NC cell migration into BA4. Similarly, results from this study show that cranial NC cells streaming in BA2 appear disrupted [5,6]. Furthermore, migratory perturbations in the post-otic cranial NC cells of *Gbx2*^−/−^ mutants are thought to contribute to the numerous pharyngeal arch artery and inner ear phenotypes observed in these mice [3].

Several lines of evidence derived from studies in *Gbx2^neo^*^/*neo*^ mutants provide additional support to the notion that *Gbx2* is required for the migration and survival of a subpopulation of trigeminal cranial NC cells: (1) the trigeminal cranial NC cell-derived sensory components, the mandibular branch, and the nV BC cells are absent; (2) neural crest EMT appeared intact, while analysis of *Sox10* in *Gbx2^neo^*^/*neo*^ mutants revealed aberrant trigeminal cranial NC cells migrating from the dorsal neural tube; (3) a loss of GBX2 target and guidance molecule, *Nrp1*, within the trigeminal crest NC cell stream; and (4) the observed increase in activated caspase-3 activity in a subpopulation of AP2α-positive trigeminal cranial NC cells and in the surrounding mesenchyme. It is unclear whether *Gbx2* is required cell-autonomously, non-cell-autonomously, or both, for the normal development of cranial NC cells.

It remains possible that the loss of tissue and multiple r2-specific genes in *Gbx2^neo^*^/*neo*^ mutants may contribute to the migratory defects and the increase in cell death observed within the trigeminal cranial NC cell stream. The increase in apoptosis observed within the migrating trigeminal cranial NC cell stream may also account for the loss of *Nrp1* expression in migrating NC cells and suggest a possible GBX2-mediated mechanism for nV development. Interestingly, an anti-apoptotic response from VEGF/NRP1 signaling has been reported in multiple cell types including cancer cells, stem cells, and neurons [66,67,68]. It is possible that the loss of the migratory factor *Nrp1,* and the overall mispatterning of the anterior hindbrain, contribute to the migration defects and the increase in the trigeminal cranial NC cell death in *Gbx2^neo^*^/*neo*^ mutants. 

The proposed functions of *Gbx2* in the patterning and survival of cranial NC cells and the expression of *Krox20* in the anterior hindbrain are similar to those observed with other homeobox-containing transcription factors. *Msx1* and *Msx2* have been implicated in the specification of cranial NC cells and are required for their migration and survival. Moreover, *Gbx2*, *Msx1*, and *Msx2* are required for normal patterning of the anterior hindbrain markers *Krox20* and *Epha4* [68].

Much of what we know about cranial NC cell migration in the anterior hindbrain in the mouse has come from studies investigating cranial NC cell streaming from r4 into BA2 and the caudal cardiac streams. However, few studies have attempted to describe the complex interplay between the molecular network of transcription factors and signaling molecules that mediate the migration and survival of the trigeminal cranial NC cells. The results presented in this study further expand our knowledge of *Gbx2* function within the gene regulatory network underlying migratory cranial NC cells contributing to nV and anterior hindbrain development. Figure 6 provides a summary of the morphological and gene expression defects observed in a lateral view of *Gbx2*^−/−^ and *Gbx2^neo^*^/*neo*^ embryonic mice compared to wild-type embryos at E9.5.

The data from studies in *Gbx2^neo^*^/*neo*^ mutant embryos revealed ectopic expression of *Krox20* in r3, loss of *Nrp1* expression in a subset of migrating trigeminal NC cells, and overall reduction of NC cells, likely due to an increase in cell death (Figure 6C).

## Figures and Tables

**Figure 1 jdb-08-00033-f001:**
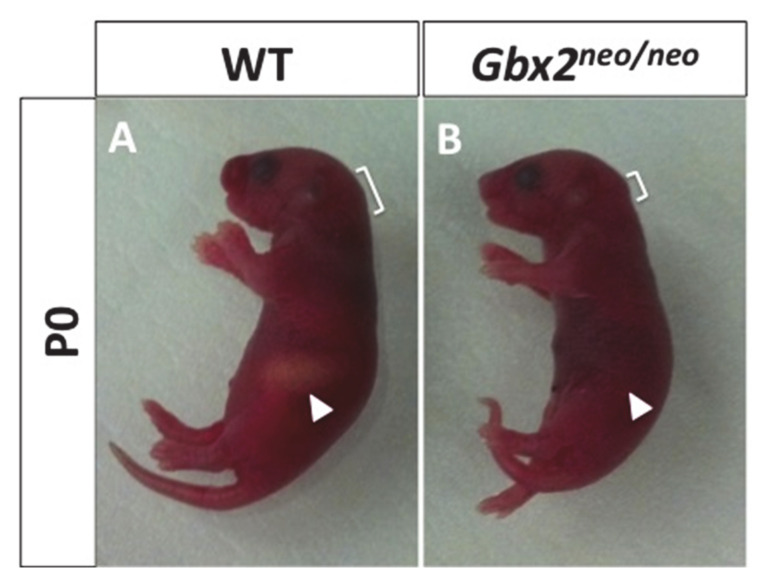
Loss of *Gbx2* results in neonatal lethality. Representative photograph of P0 wild-type (**A**) and *Gbx2^neo^*^/*neo*^ mice (**B**). *Gbx2^neo^*^/*neo*^ lack milk in their stomachs compared to wild-type mice (white arrowhead).

**Figure 2 jdb-08-00033-f002:**
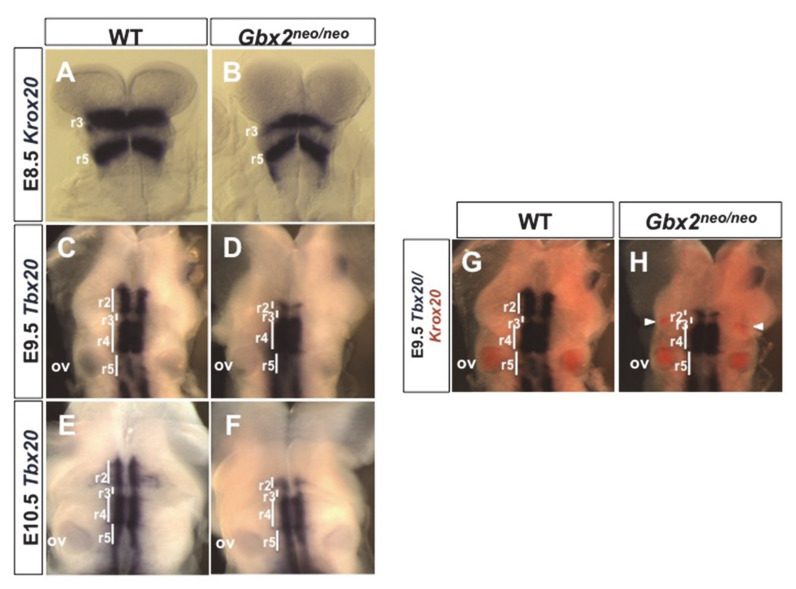
Developmental defects in the trigeminal motor neurons in *Gbx2^neo^*^/*neo*^ embryos. Whole and flat-mounted in situ hybridization analyses of *Krox20* and *Tbx20* in anterior hindbrain at E8.5, and E9.5 and E10.5. Whole-mount preparation (**A**,**B**). Dorsal view of *Krox20* expression in wild-type (**A**) and *Gbx2^neo^*^/*neo*^ (**B**) embryos at E8.5. (**C**–**F**) Flat-mounted hindbrain preparations at E9.5 and E10.5. Dorsal view of *Tbx20* expression in the ventral motor columns in wild-type (**C**,**E**) and *Gbx2^neo^*^/*neo*^ (**D**,**F**) embryos. *Tbx20* expression is clearly observed in r2–r5 motor neurons in wild-type controls while *Tbx20* expression appears to be significantly reduced in the r2 motor neurons in *Gbx2^neo^*^/*neo*^ embryos at E9.5 and E10.5. (**G**,**H**) Flat-mounted hindbrain preparations at E9.5. *Krox20* expression is downregulated in r3 at E9.5 in wild-type embryos (**G**). Ectopic *Krox20* expression persist in r3 in *Gbx2^neo^*^/*neo*^ embryos (**H**). r, rhombomere; ov, otic vesicle; white bar, represents span of adjacent rhombomere, r.

**Figure 3 jdb-08-00033-f003:**
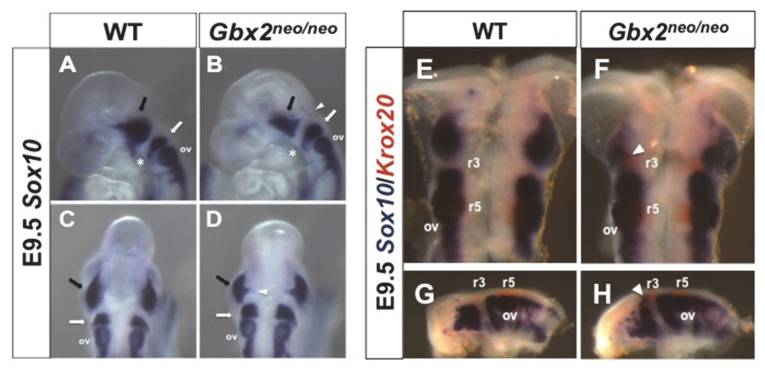
Trigeminal cranial NC cells fail to migrate into BA1 and remain adjacent to the dorsal neural tube in *Gbx2^neo^*^/*neo*^ embryos. Whole-mount in situ hybridization analysis of *Sox10* at E9.5 in lateral views (**A**,**B**) and dorsal views (**C**,**D**) demonstrate that trigeminal cranial NC cells fail to migrate into BA1 and remain adjacent to the dorsal neural tube in *Gbx2^neo^*^/*neo*^ mutants. Image analysis of *Sox10* reveals that trigeminal cranial NC cells in *Gbx2^neo^*^/*neo*^ embryos fail to migrate into BA1 compared to wild-type controls (**A**,**B**). *Sox10* is abnormally expressed (white arrowhead) as a lateral stripe (**B**) and additional punctate expression (**D**) in the dorsal neural tube within the trigeminal cranial NC cell stream (black arrows) in *Gbx2^neo^*^/*neo*^ embryos compared to wild-type controls (**A**,**C**). *Sox10* expression in the hyoid cranial NC cell population (white arrows) and the otic vesicle appears unaffected in *Gbx2^neo^*^/*neo*^ mutants. Two-color in situ hybridization analysis of *Sox10* and *Krox20* on flat-mounted hindbrain preparations in wild-type (**E**,**G**) and *Gbx2^neo^*^/*neo*^ (**F**,**H**) embryos at E9.5 show that ectopic *Krox20*-expressing cells observed in *Gbx2^neo^*^/*neo*^ mutants are not cranial NC cells. *Sox10* expression in dorsal (**E**,**F**) or lateral views (**G**,**H**) is anterior (white arrowhead) to the *Krox20* expression in r3 in *Gbx2^neo^*^/*neo*^ embryos. r, rhombomere; ov, otic vesicle; asterisk, BA1.

**Figure 4 jdb-08-00033-f004:**
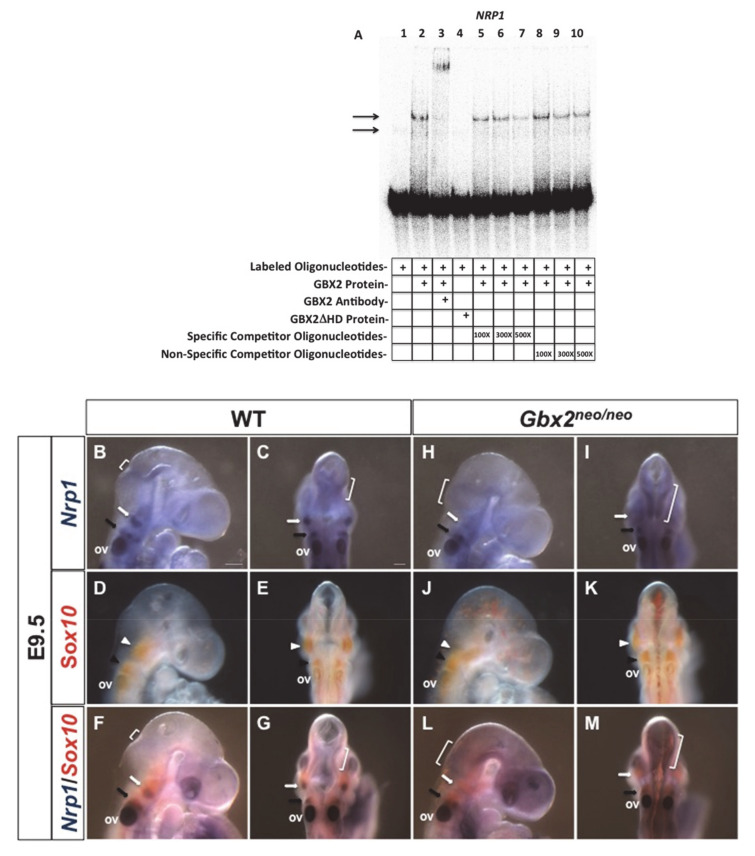
GBX2 binding to *NRP1* and loss of *Nrp1* in subpopulation of migrating trigeminal cranial NC cells in *Gbx2^neo^*^/*neo*^ embryos. (**A**) Gel-shift analysis for GBX2 target gene, *NRP1*. A reduction in the mobility of (γ-32P) ATP-labeled *NRP1* probe is observed with the addition of GBX2 (black arrows, lane 2). A supershift is observed with the addition of a GBX2 antibody (lane 3), whereas no shift is observed with the addition of GBX2ΔHD (lane 4). Addition of unlabeled specific competitor oligonucleotides (lanes 5–7) and 45-mer non-specific competitor oligonucleotides (lanes 8–10), omitting the GBX2 DNA-binding sequence, at 100×, 300×, and 500× molar concentrations. Whole-mount in situ hybridization analysis for *Nrp1* (**B**,**C**,**H**,**I**) and *Sox10* (**D**,**E**,**J**,**K**) and two-color in situ hybridization (**F**,**G**) in wild-type and (**L**,**M**) in *Gbx2^neo^*^/*neo*^ embryos at E9.5. Lateral and dorsal views show *Nrp1* expression in two distinct populations of cells (**B**,**C**, arrows) corresponding to *Sox10* expressing trigeminal (white arrowhead) and hyoid (black arrowhead) cranial NC cells in wild-type embryos (**D**,**E**) and *Gbx2^neo^*^/*neo*^ mutants (**J**,**K**). Two-color in situ analysis for *Nrp1* and *Sox10* reveals a reduction in *Nrp1* expression within a subpopulation of *Sox10*-positive trigeminal cranial NC cells in *Gbx2^neo^*^/*neo*^ embryos (compare white arrows in (**L**) and (**M**) with (**F**) and (**G**)) while *Nrp1* expression in the hyoid cranial NC cell stream and the otic vesicle appears unaffected. Brackets indicate ectopic *Nrp1* expression throughout r1 in *Gbx2^neo^*^/*neo*^ embryos compared to wild-type controls. ov, otic vesicle.

**Figure 5 jdb-08-00033-f005:**
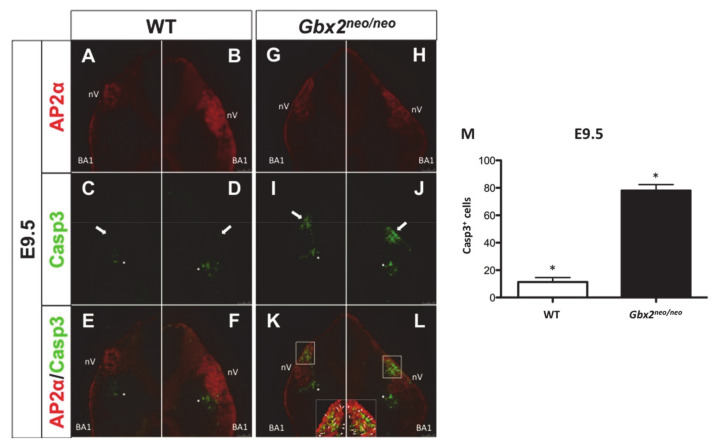
An increase in cell death is observed in *Gbx2^neo^*^/*neo*^ trigeminal cranial NC cells. (**A**–**L**) Immunofluorescence staining of transverse sections through nV using AP2α and the apoptotic marker, caspase-3 (Casp3), in E9.5 embryos identified an increase in cell death in cranial NC cells and the surrounding mesenchyme in *Gbx2^neo^*^/*neo*^ mutants (compare white arrows in (**C**,**D**) to (**I**,**J**). Casp3 positive cells are observed in the mesenchyme medial to nV (*) in both wild-type (**C**–**F**) and *Gbx2^neo^*^/*neo*^ embryos (**I**–**L**). A dramatic increase in Casp3 positive cells colocalizing with AP2α in migrating cranial NC cells (inset, white arrows), and an increase in Casp3 in the surrounding mesenchyme (inset, white arrowheads) dorsal of the nV ganglion, is observed in *Gbx2^neo^*^/*neo*^ mutants (**K**,**L**) compared to wild-type controls (**E**,**F**). Increase in the number Casp3 positive cells in *Gbx2^neo^*^/*neo*^ mutants compared to wild-type controls (**M**).

**Figure 6 jdb-08-00033-f006:**
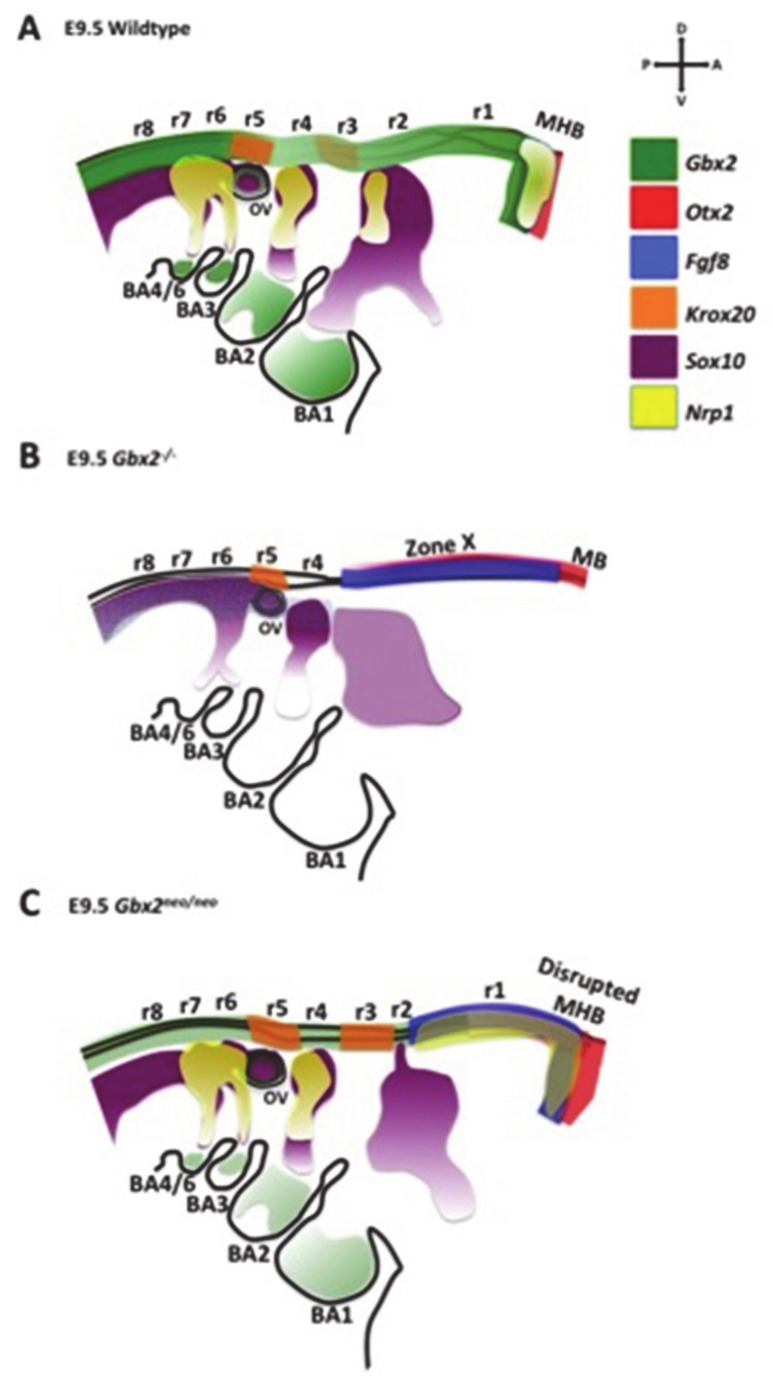
Representation of the morphological and gene expression defects observed in a lateral view of *Gbx2*^−/−^ (**B**) and *Gbx2^neo^*^/*neo*^ embryonic mice (**C**) compared to wild-type embryos (**A**) at E9.5. Colored areas represent the gene expression domains as indicated by the key. A loss in color intensity represents a reduction in the expression of the corresponding gene compared to the wild-type embryo. (**B**) Zone X indicates the loss of r1–r3 and the abnormal gene expression within the *Gbx2*^−/−^ embryo. (**C**) The disrupted MHB in the *Gbx2^neo^*^/*neo*^ embryos indicates the caudal expansion of gene expression that is normally restricted to this region.

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
