# Peer review of "Gbx2* Is Required for the Migration and Survival of a Subpopulation of Trigeminal Cranial Neural Crest Cells"

_jdb, 2020, doi:10.3390/jdb8040033_

Round 1

Reviewer 1 Report

In this study Roeseler et al investigate the role of Gbx2 in anterior hindbrain development focusing on cranial Neural Crest Cells (cNCC) and motor neurons migration. They try to improve understanding of mechanisms regulated by Gbx2 in anterior hindbrain, and trigeminal nerve development using anterior hindbrain development in the hypomorph mutant of Gbx2: Gbx2neo/neo. Using Tbx1 as a marker they observed a reduction of rhombomere 2 (r2) size and supposedly of r2-derived motor neurons presence. They also described changes in Krox20 temporal expression with a late ectopic expression in rhombomere 3. Finally the author observe that Gbx2 is required for the expression of Neuropilin 1 (Nrp1) in a subpopulation of trigeminal NC cells and an increase of apoptosis in cNCC in these mice.

All together there are some interesting observations in this manuscript however it needs serious revision before being accepted for publication.

First and foremost an intensive rewriting is necessary. At this point the paper is verbose and very difficult to read, evaluate and appreciate. It is still unclear to me what is new in this study compare to previous ones, specifically those performed on Gbx2 knock out mice. One example is the description in the introduction of retinoids, Fgf and Wnt activities in the hindbrain, which is irrelevant to the results, whereas a clear description of Gbx2 knock out phenotype in mouse and/or other species in the anterior hindbrain is required and absent.

In the discussion section the differences between Gbx2-/- and Gbx2neo/neo phenotypes is necessary and absent.

There are numerous observations, which are presented but neither discussed not connected to the point the authors are trying to make which is very confusing.

For example:

Figure 1: the authors describe clear sucking defect in their mice, and never use this information nor talked about it in the discussion

Figure 2A: What is the Krox20 expression at E8.5 bringing to the argument?

Figure 3 Lane 241

What is the interest of showing that Krox20 expressing cells located in r3 are not Neural Crest cells?

The writing is not precise and thereby confusing

Lane 157 “arise from r2:r3” means r2 and r3, or the boundary between r2 and r3.

Lane 163 what does the authors mean “the status of r3”?

Lane 184 I do not understand the sentence “In contrast to… (figure 2D F)”; The two results seem to go in the same direction.

Some results are not convincing

Figure 3E-L and Lanes 232 to 234:

The downregulation of E cadherin is very difficult to assess. I cannot make any sense of what I am observing. I am not seeing a down-regulation of E cadherin or maybe in Figure 3I and 3K.

Besides in Figure 3 what is the difference between E versus F, I versus J, G versus H, and K versus L?

To make their point the authors have either to provide other images, or find another way to demonstrate that EMT is not affected in Gbx2 neo/neo mice.

Figure 4

Whereas the arguments that Gbx2 protein binding to Gbx2 CRM in Nrp1 sequence, and the marked reduction of Nrp1 in r2, are reasonably convincing the extension into r1 is difficult to apprehend and deserve no more then a note. Especially I do not see the relevance to the point.

Figure 5

Again what is the difference between A versus B, G versus H, C versus D, and I versus J?

I observe differences between A and B and I dot not know what I am looking at.

Whereas the increase in apoptosis is observable on Figure 5 the phenotype quantification is required.

The results are confusing. A scheme of the most important results is necessary as a description of the novelty of these results

Author Response

November 24, 2020

Comments and Suggestions for Authors

In this study Roeseler et al investigate the role of Gbx2 in anterior hindbrain development focusing on cranial Neural Crest Cells (cNCC) and motor neurons migration. They try to improve understanding of mechanisms regulated by Gbx2 in anterior hindbrain, and trigeminal nerve development using anterior hindbrain development in the hypomorph mutant of Gbx2: Gbx2neo/neo. Using Tbx1 as a marker they observed a reduction of rhombomere 2 (r2) size and supposedly of r2-derived motor neurons presence. They also described changes in Krox20 temporal expression with a late ectopic expression in rhombomere 3. Finally the author observe that Gbx2 is required for the expression of Neuropilin 1 (Nrp1) in a subpopulation of trigeminal NC cells and an increase of apoptosis in cNCC in these mice.

All together there are some interesting observations in this manuscript however it needs serious revision before being accepted for publication.

First and foremost an intensive rewriting is necessary. At this point the paper is verbose and very difficult to read, evaluate and appreciate. It is still unclear to me what is new in this study compare to previous ones, specifically those performed on Gbx2 knock out mice. One example is the description in the introduction of retinoids, Fgf and Wnt activities in the hindbrain, which is irrelevant to the results, whereas a clear description of Gbx2 knock out phenotype in mouse and/or other species in the anterior hindbrain is required and absent.

We have reduced the content of the manuscript. We have specifically reduced content regarding Fgf and Wnt. activities in the hindbrain. We have re removed lines 50 through line 62 as requested by the reviewer. We have provided description of Gbx2 knockout phenotypes.

In the discussion section the differences between Gbx2-/- and Gbx2neo/neo phenotypes is necessary and absent.

We have provided discussion of Gbx2-/- and Gbx2neo/neo phenotypes in the Introduction and Discussion.

There are numerous observations, which are presented but neither discussed not connected to the point the authors are trying to make which is very confusing.

For example:

Figure 1: the authors describe clear sucking defect in their mice, and never use this information nor talked about it in the discussion

We have discussed the suckling defect and relevance in the Discussion.

Figure 2A: What is the Krox20 expression at E8.5 bringing to the argument?

Figure 3 Lane 241

What is the interest of showing that Krox20 expressing cells located in r3 are not Neural Crest cells?

We have expanded the discussion on Krox20 to include the relevance of them being Neural Crest or not.

The writing is not precise and thereby confusing

Lane 157 “arise from r2:r3” means r2 and r3, or the boundary between r2 and r3.

We have clarified the meaning

Lane 163 what does the authors mean “the status of r3”?

We have clarified the meaning

Lane 184 I do not understand the sentence “In contrast to… (figure 2D F)”; The two results seem to go in the same direction.

We have clarified the meaning

Some results are not convincing

Figure 3E-L and Lanes 232 to 234:

The downregulation of E cadherin is very difficult to assess. I cannot make any sense of what I am observing. I am not seeing a down-regulation of E cadherin or maybe in Figure 3I and 3K.

Besides in Figure 3 what is the difference between E versus F, I versus J, G versus H, and K versus L?

To make their point the authors have either to provide other images, or find another way to demonstrate that EMT is not affected in Gbx2 neo/neo mice.

We agree with the reviewer that the data are not overly convincing and further experiments are required to demonstrate if EMT is affected in Gbx2 neo/neo embryos. Therefore, we removed this section of the manuscript.

Figure 4

Whereas the arguments that Gbx2 protein binding to Gbx2 CRM in Nrp1 sequence, and the marked reduction of Nrp1 in r2, are reasonably convincing the extension into r1 is difficult to apprehend and deserve no more then a note. Especially I do not see the relevance to the point.

We have minimized the discussion regarding expansion of NRP1 expression into r1.

Figure 5

Again what is the difference between A versus B, G versus H, C versus D, and I versus J?

We have provided explanations for each panel in Figure 5.

I observe differences between A and B and I dot not know what I am looking at.

Panels A and B are showing expression of AP2 alpha to indicate migrating NC cells in NV. Likewise with Panels G and H. A comparison between the two panels is not needed

Whereas the increase in apoptosis is observable on Figure 5 the phenotype quantification is required.

We have counted and graphed the Casp3-positive cells to show the difference between WT and Gbx2 neo/neo embryos.

The results are confusing. A scheme of the most important results is necessary as a description of the novelty of these results

We have provided a new figure summarizing the results.

Reviewer 2 Report

The study by Roseler, et al. describe the investigation into the phenotype of a Gbx2 hypomorph mouse. The authors describe interesting findings related to cranial neural crest and GBX2 and changes in rhombomere 2.

Minor comments:

1) It was unclear in the beginning how this Gbx2-Neo strain was different from the original Gbx2-deltaHB mouse in Wassarman et al (1997). This as due to the fact that the Gbx2-deltaHB mouse still contains the Neo cassette and Wassarman refer to the variant used in this study as the Gbx2-floxed mouse. It would help if some additional information could be added to the methods section that would clearly state that this mouse still contains all exons and the hypomerphic phenotype is due to the inclusion of the neo cassette.

2) Figure 2, the white bars are not explained in the figure legend.

Overall, the manuscript is well-planned and it presents some interesting and novel results. In my opinion, the manuscript is suitable for publication once these minor points are addressed.

Author Response

November 24, 2020

Listed below are our responses to reviewer comments

Reviewer 2:

Comments and Suggestions for Authors

The study by Roseler, et al. describe the investigation into the phenotype of a Gbx2 hypomorph mouse. The authors describe interesting findings related to cranial neural crest and GBX2 and changes in rhombomere 2.

Minor comments:

1) It was unclear in the beginning how this Gbx2-Neo strain was different from the original Gbx2-deltaHB mouse in Wassarman et al (1997). This as due to the fact that the Gbx2-deltaHB mouse still contains the Neo cassette and Wassarman refer to the variant used in this study as the Gbx2-floxed mouse.

It would help if some additional information could be added to the methods section that would clearly state that this mouse still contains all exons and the hypomerphic phenotype is due to the inclusion of the neo cassette.

We have provided language in the Materials and Methods section describing the underlying mechanism contributing to the hypomorphic phenotype of the Gbx2-Neo strain.

2) Figure 2, the white bars are not explained in the figure legend.

We have provided explanation for the white bars in the figure legend.

Overall, the manuscript is well-planned and it presents some interesting and novel results. In my opinion, the manuscript is suitable for publication once these minor points are addressed.

Round 2

Reviewer 1 Report

The authors have answered in a satisfactory way to my concerns.
Minor point

Lane 332 what is the sign between AP2a and positive?

There are changes of font in different parts of the manuscript (lane 496, 336, 337, 661 etc…)

The scheme presented in figure 6 is a great adjunct.

Figure 6 I noted that a loss in colour intensity represents a reduction in the expression of the corresponding gene compared to the wild-type. However I did not understand what represents the brownish yellow colour in the area where the Neural Crest cells normally migrate.

Author Response

December 06, 2020

Comments and Suggestions for Authors

The authors have answered in a satisfactory way to my concerns.
Minor point

Lane 332 what is the sign between AP2a and positive?

We have corrected the sign between AP2a and positive.

There are changes of font in different parts of the manuscript (lane 496, 336, 337, 661 etc…)

We have examined the manuscript and corrected typos and font

The scheme presented in figure 6 is a great adjunct.

Figure 6 I noted that a loss in colour intensity represents a reduction in the expression of the corresponding gene compared to the wild-type. However I did not understand what represents the brownish yellow colour in the area where the Neural Crest cells normally migrate.

We have corrected (removed) the brownish yellow colour in the area where the Neural Crest cells normally migrate in Figure 6C.